# Attenuation of Lipopolysaccharide-Induced Acute Lung Injury by Hispolon in Mice, Through Regulating the TLR4/PI3K/Akt/mTOR and Keap1/Nrf2/HO-1 Pathways, and Suppressing Oxidative Stress-Mediated ER Stress-Induced Apoptosis and Autophagy

**DOI:** 10.3390/nu12061742

**Published:** 2020-06-10

**Authors:** Ching-Ying Huang, Jeng-Shyan Deng, Wen-Chin Huang, Wen-Ping Jiang, Guan-Jhong Huang

**Affiliations:** 1Graduate Institute of Aging Medicine, School of Medicine, China Medical University, Taichung 404, Taiwan; bryan1479@gmail.com; 2Department of Health and Nutrition Biotechnology, Asia University, Taichung 413, Taiwan; dengjs@asia.edu.tw; 3Graduate Institute of Biomedical Sciences, School of Medicine, China Medical University, Taichung 404, Taiwan; huangwc@mail.cmu.edu.tw; 4Department of Chinese Pharmaceutical Sciences and Chinese Medicine Resources, College of Chinese Medicine, China Medical University, Taichung 404, Taiwan; center.pan@gmail.com

**Keywords:** hispolon, anti-inflammation, apoptosis, LPS, HO-1, Nrf-2, ER stress

## Abstract

The anti-inflammatory effect of hispolon has identified it as one of the most important compounds from *Sanghuangporus sanghuang*. The research objectives were to study this compound using an animal model by lipopolysaccharide (LPS)-induced acute lung injury. Hispolon treatment reduced the production of the pro-inflammatory mediator NO, TNF-α, IL-1β, and IL-6 induced by LPS challenge in the lung tissues, as well as decreasing their histological alterations and protein content. Total cell number was also reduced in the bronchoalveolar lavage fluid (BALF). Moreover, hispolon inhibited iNOS, COX-2 and IκB-α and phosphorylated IKK and MAPK, while increasing catalase, SOD, GPx, TLR4, AKT, HO-1, Nrf-2, Keap1 and PPARγ expression, after LPS challenge. It also regulated apoptosis, ER stress and the autophagy signal transduction pathway. The results of this study show that hispolon regulates LPS-induced ER stress (increasing CHOP, PERK, IRE1, ATF6 and GRP78 protein expression), apoptosis (decreasing caspase-3 and Bax and increasing Bcl-2 expression) and autophagy (reducing LC3 I/II and Beclin-1 expression). This in vivo experimental study suggests that hispolon suppresses the LPS-induced activation of inflammatory pathways, oxidative injury, ER stress, apoptosis and autophagy and has the potential to be used therapeutically in major anterior segment lung diseases.

## 1. Introduction

Acute lung injury (ALI) is the leading cause of death in patients with respiratory failure. It may occur through direct injury, such as pneumonia and the inhalation of harmful substances, or through indirect damage, such as ischemia-reperfusion, the aspiration of gastric contents, sepsis and multiple injuries or acute pancreatitis, which may cause an unstable redox state that can cause DNA damage or protein and lipid oxidation [1]. ALI can induce damaging hypoxia and ischemic stress or the release of bacterial endotoxins, such as lipopolysaccharide (LPS) [2]. LPS can induce a serious release of inflammatory signals and cause lung damage characterized by inflammatory leukocyte infiltration. The severity of the pneumonia is caused by the great accumulation of neutrophils, increased reactive oxygen species (ROS) production, and enhanced proinflammatory cytokines production, such as tumor necrosis factor (TNF-α), interleukin-1β (IL-1β) and IL-6, which exacerbate the inflammation, leading to secondary diffuse lung parenchymal damage in the alveoli [3]. Therefore, it is important to find an agent with the ability to inhibit the inflammatory response for the treatment of ALI.

The activation of the Toll-like receptor 4 (TLR4) signaling pathway leads to the production of inflammatory mediators, which results in the translocation of nuclear factor-κB (NF-κB) and the activation of mitogen-activated protein kinase (MAPK), which plays a key role in triggering inflammation after the LPS challenge [4]. Phosphatidylinositol-3 kinase/protein kinase B/mammalian target of rapamycin (PI3K/Akt/mTOR) signaling has been proposed as a significant functional regulator of TLR4/NF-κB activity [5]. It can promote cell division and proliferation and plays key roles in suppressing apoptosis, immune regulation [6], regulating anti-inflammatory cytokines and promoting the degradation of the extracellular matrix [7].

Oxidative stress-mediated injury plays an essential role in lung failure. The protective mechanism is currently believed to be derived from antioxidant enzymes (catalase, superoxide dismutase (SOD) and glutathione peroxidase (GPx)) and nuclear factor erythroid 2-related factor (Nrf2)/heme oxygenase (HO-1) signaling against oxidative stress [8]. Nrf2 is an inducible protein with cytoprotective and antioxidant activities. Under normal circumstances, Nrf2 binds to Kelch-like ECH-related protein 1 (Keap1) in the cytosol. When there is a redox imbalance in the cell, free Nrf2 binds antioxidant response elements in the nucleus, thereby activating detoxification gene expression [9]. HO-1 catalyzes the breakdown of heme, thereby inhibiting neutrophil-, macrophage- and lymphocyte-mediated inflammatory responses [10]. Oxidative stress and certain pro-inflammatory mediators can induce Nrf2 activation and HO-1 expression. HO-1 is a cytoprotective protein produced through the Keap1/Nrf2/HO-1 axis and can curtail the cytotoxicity of various sources of oxidative stress and inflammatory reactions [11].

The endoplasmic reticulum (ER) serves many roles in the cell, including in protein folding, transportation, cellular calcium storage, lipid and steroid synthesis, and the metabolism of carbohydrates [12,13]. Under cellular stress and inflammatory conditions, ER stress activates the unfolded protein response (UPR) to relieve this stress and restore ER homeostasis [14]. Under pathological conditions (such as ALI, sepsis and infection), ER stress may lead to the accumulation of a large number of incorrect protein structures [15,16]. Three ER transmembrane proteins are affected by ER stress: protein kinase R–like ER kinase (PERK), inositol-requiring kinase 1 (IRE1), and activating transcription factor 6 (ATF6). Glucose-regulated protein 78 (GRP78, also known as BiP) binds to IRE1, PERK and ATF6 in stress-free cells and dissociates from these UPR sensors during ER stress [17]. C/EBP homologous protein (CHOP) and caspase-12 are also key molecules in ER-induced apoptosis. Severe and prolonged ER stress strongly induces CHOP [14]. Additionally, caspase-12 induces the downstream death molecule caspase-3 when activated by ER stress [18]. Thus, ER stress potentially affects survival in ALI [16].

Autophagy maintains energy homeostasis, allowing cells to survive under nutrient restriction [19]. Two signaling molecules tightly control the autophagy-mediated activation of AMP-activated protein kinase (AMPK), which induces autophagy and the mammalian target of rapamycin (mTOR) via the autophagy axis [20]. Autophagy is achieved through the formation of autophagosomes, which are involved in the conversion of light chain 3-I (LC3-I) to LC3-II in the cytosol. Therefore, the ratio of LC3-I to LC3-II can be used as autophagy markers [21]. Beclin 1 plays a key role in the localization of other autophagic proteins to the autophagosome prestructure. Thus, LC3-II and beclin 1 are often measured as autophagic markers [22]. However, little is known about how autophagy is regulated during ALI development.

AMPK plays a central role in the control of energy balance and can sense the energy state of cells by detecting the amount of AMP. Its activation enhances energy production through liver kinase B1 (LKB1) and calcium/calmodulin-dependent protein kinase 2 (CaMKK2) [23]. ER calcium release contributes to an increase in intracellular calcium, which causes the production of ROS and the activation of AMPK. Reports suggest that LKB1/CaMKK/AMPK signaling may promote lung tissue injury [24].

Hispolon is a natural bioactive polyphenol from *S. sanghuang* or *Phellinus linteus* used as a herbal medicine in Taiwan, Korea, China and Japan [25]. Our previous publication indicates that hispolon protects the livers against CCl4-induced acute liver injury [26]. Other studies have shown antitumor actions for hispolon against human leukemia [27] and liver cancer [28] and human hepatoma cell metastasis [29]. Hispolon can induce the protein expression of HO-1 and inhibit iNOS (inducible nitric oxide synthase) and NO (nitric oxide) productions in LPS or LTA-induced BV-2 cells [30]. Although many studies have demonstrated the anti-inflammatory potential of hispolon, the mechanism of hispolon protecting against LPS-induced ALI has not been evaluated. Therefore, we hypothesize that the TLR4/PI3K/Akt/mTOR and Keap1/Nrf2/HO-1 signalings, the modulation of oxidative stress and the ER stress-induced apoptosis axis involve the anti-inflammation of hispolon in LPS challenge ALI. Thus, the purpose of this research was to investigate the antioxidative, anti-inflammatory, and anti-apoptotic functions of hispolon.

## 2. Materials and Methods

### 2.1. Chemicals and Reagents

Hispolon was acquired from BJYM Pharmaceutical. & Chemical Co., Ltd. (Beijing, China). The purity of hispolon was higher than 95% (Figure 1A). LPS, dexamethasone (DEX) and other chemicals, solvents, and reagents were obtained from Sigma-Aldrich (St. Louis, MO, USA). Enzyme-linked immunosorbent assay (ELISA) kits for the determination of cytokine secretion were acquired from BioLegend Inc. (San Diego, CA, USA). Anti-PI3k and Anti-p-AKT were obtained from Merck Millipore (Merck KGaA, Darmstadt, Germany). The antibodies against TLR4, AKT, p-JNK, ATF6, p-CaMKK2, p-LKB1, catalase, GPx, SOD, Keap1, COX-2, caspase 12, IRE1, GRP78, PERK, CHOP, Beclin 1 and LC3 I/II were obtained from GeneTex (Irvine, CA, USA). Antibodies against IKK, p-IKK, JNK, p-ERK, ERK, p-p38, mTOR and p-mTOR were purchased from Cell Signaling Technology (Beverly, MA, USA). Anti-iNOS, anti-HO-1, anti-Nrf-2, anti-PPARγ, anti-IκBα, anti-NF-κB, anti-p38, and anti-β-actin were purchased from Abcam (Cambridge, UK). Determination of protein concentration using a Bio-Rad protein assay kit (Bio-Rad Laboratories Ltd., Watford, UK).

### 2.2. Animals

Six-week-old male ICR mice (25–28 g) were obtained from BioLASCO Taiwan Co., Ltd. (Taipei, Taiwan). Animal testing was conducted in accordance with the guidelines set out by the China Medical University Animal Care Committee (IACUC approval number: 104-127-N).

### 2.3. Experimental Protocols

After a minimum of 7 days of adaptation, mice were randomly divided into six treatment groups (n = 6): control group, LPS-treated group (5 mg/kg dissolved in sterile saline, *ip*), LPS + Dex group (10 mg/kg, *ip*) and LPS + hispolon group (2.5, 5 and 10 mg/kg suspended in 0.5% CMC solution, *ip*). Intratracheal instillation of LPS induces ALI, then one hour before LPS administration, hispolon and Dex were injected intraperitoneally. Mice received sterile saline (50 μL, control group). The dose of hispolon was selected according to a previous study. After 6 h, the animals were sacrificed and samples were collected [31].

### 2.4. BALF Collection and Total Cell Counts from Mice

The bronchoalveolar lavage fluid (BALF) of each individual mouse was collected by lavaging the lung three times with saline, so that supernatants could then be collected for later analysis by ELISA and protein study. Each cell pellet was suspended in saline to calculate the total number of cells using a hemocytometer. The sediment was re-suspended to determine the total number of cells and protein content.

### 2.5. Nitrite Assay

The determination of BALF nitrite level was measured using Griess reagent [31]. Briefly, we added equal volumes of Griess reagent and BALF solution and mixed. After incubating 10 min at room temperature, the absorbance value was recorded at 540 nm, using an ELISA Plate Reader.

### 2.6. Histopathological Analysis

The right lower pulmonary tissue was removed for pulmonary histopathology. The pulmonary tissue was placed in 4% paraformaldehyde, then embedded in paraffin. Hematoxylin and eosin (H&E) stains were carried out using a standard procedure and photographed with a light microscope. The severity of lung injury was scored from 0 to 5, depending on the degree of inflammatory cell infiltration (including neutrophils) and dissemination. A score of 0 indicates a normal state; 1 represents a minimum (<1%), 2 represents a slight (1–25%), 3 represents a mild (26–50%), 4 represents a severe (51–75%) and 5 represents an excessive expression of lung injury (76–100%) [32].

### 2.7. ELISA Analysis in the BALF

The levels of serum pro-inflammatory cytokines in BALF were determined by enzyme-linked immunasorbent (ELISA) kits, respectively.

### 2.8. Pulmonary Tissue Wet/Dry Weight Ratio

Scales were used to evaluate the rate of changes of wet/dry (W/D) ratio after different treatments. The left lower lobe was dried and weighed, then placed in an incubator for 48 h at 60 °C, to obtain the dry weight. The calculation of the W/D ratio can assess the degree of lung tissue edema.

### 2.9. Myeloperoxidase Activity

Tissues were homogenized in sterile saline. The homogenate was centrifuged and the pelleted was suspended pellet in buffer (50 mM K_2_HPO_4_, pH 6.0) containing 0.19 mg/mL of o-dianisidine chloride and 0.0005% hydrogen peroxide as a substrate [33]. Spectral absorption was measured at 460 nm. Myeloperoxidase (MPO) activity was measured as the optical density (OD) at 460 nm/mg protein.

### 2.10. Determination of ROS

The BALF was incubated with 100 μM 2′,7′-dichlorofluorescein-diacetate (DCFH-DA) in the dark, at room temperature for 30 min. The fluorescence intensity was measured with an excitation wavelength at 485 nm and emission wavelength at 535 nm, using the Synergy Fluorescence Intensity HTMicroplate Reader (BioTek Instruments, Winooski, VT, USA) [9].

### 2.11. Western Blot Analysis

Lung tissue was homogenized by using RIPA lysis buffer containing protease inhibitors, then centrifuged to measure total protein. Equal amounts of protein samples were loaded onto the gels SDS–PAGE gel and electrophoresed. Following electrophoresis, proteins were transferred onto a solid support (PVDF membrane), which was then blocked and incubated with primary antibody at an appropriate dilution in PBS, which was conjugated to horseradish peroxidase (HRP) and detected using the enhanced chemiluminescent method. Quantify the bands using densitometry (Eastman Kodak Company, Rochester, NY, USA), respectively.

### 2.12. Statistical Analyses

Data were presented as mean ± standard error of the mean (SEM). Differences among multiple groups were analyzed by a one-way analysis of variance (ANOVA) and Student’s *t*-test. ^###^ indicates *p* < 0.001 compared with the control group; * indicates *p* < 0.05, ** indicates *p* < 0.01, and *** indicates *p* < 0.001, compared to the LPS alone group.

## 3. Results

### 3.1. Hispolon Decreases LPS-Induced Histopathological Changes in the Mouse Lung

Morphological changes in lungs following the LPS challenge were examined. The results showed normal lung architecture in the control group, while pulmonary vessels infiltrated by neutrophils and edema in the interstitial space of the alveolar wall were observed in the LPS group, indicative of alveolar epithelial cell damage. These pathological processes could be ameliorated by hispolon and Dex in mice, indicating that hispolon alleviated the pathological effects in the LPS-challenged ALI mouse model (Figure 1B,C). Moreover, the pulmonary injury score displayed that hispolon inhibited the LPS challenge inflammatory response against the histopathological changes, in this LPS-induced ALI mouse model.

### 3.2. Decreased Pulmonary W/D Weight Ratio and MPO Activity

Lung edema and vascular permeability were increased with LPS induction compared to the control, as indicated by the altered pulmonary W/D ratio. However, hispolon and Dex treatment reduced the W/D ratio in the lung, compared to the mice treated with LPS-instilled alone group (Figure 2A). These findings indicate that hispolon can protect against LPS challenge-induced pulmonary edema and lung inflammation.

Pulmonary tissues of MPO activity, a useful biomarker of neutrophil influx into the BALF and lung tissue, was measured to assess neutrophil accumulation and oxidation, which can increase inflammation and cell damage [32]. As shown in Figure 2B, the intratracheal instillation of LPS into mouse lung tissue causes a significant increase in MPO activity. When the mice were pretreated with hispolon and Dex, MPO activity was decreased, compared to the LPS-instilled alone group. These preliminary data show that hispolon prevents LPS-induced pulmonary edema and lung tissue infiltration in mice.

### 3.3. Decreased Total Cell Numbers and Protein Concentration

LPS-induced mice exhibited a significant increase in total cell numbers compared to the control group. However, hispolon or Dex significantly reduced total cell counts compared to in LPS-only mice (Figure 2C). Additionally, the total protein level decreased significantly after pretreatment with hispolon and Dex, compared with control in the BALF (Figure 2D). This shows that the inhibition of ALI by hispolon is associated with a decreased leukocyte number and inflammatory response in lung tissue.

### 3.4. Decreased Pro-Inflammatory Cytokine Levels

Examining inflammatory mediator levels by ELISA demonstrates that compared to the control, LPS-stimulated ALI mice had significantly higher expressions of NO, TNF-α, IL-1β and IL-6 in the BALF (Figure 3A–D). Hispolon and Dex treatment decreased pro-Inflammatory cytokine production after LPS induction.

### 3.5. Suppression of LPS-challenged ALI iNOS, COX-2 and IKK/IκBα/NF-κB signaling in the Lung Tissues

We examined whether pretreatment with hispolon would inhibit iNOS and cyclooxygenase-2 (COX-2) protein expression in LPS-challenged mice (Figure 4A), and our results confirmed this to be the case.

Nuclear factor κB inhibitor kinase (IKK) is known to be involved in NF-κB activation and transcription. Certain signaling that cause to the accumulation of NF-κB can be activated into the nucleus by pro-inflammatory cytokines. We tested whether pretreatment with hispolon inhibited the phosphorylation of IKK, the inhibitor of NF-κBα (IκBα), and NF-κB degradation in LPS-challenged mice (Figure 4B). The results show that hispolon regulates the induction of IKK/IκBα/NF-κB signaling by LPS.

### 3.6. Suppression of MAPK Pathway Activation

A correlation exists between the activation of MAPKs and control of inflammatory response. We found that LPS triggered the significant activation of the phosphorylation of MAPKs in the lung tissues of mice (Figure 4C). Moreover, the expression of phosphorylated MAPKs could be blunted by pretreating mice with hispolon and Dex. These data show that hispolon downregulated the MAPK signaling pathway after LPS challenge.

### 3.7. The Activation of LPS-Induced Antioxidant Enzymes and PPARγ Signaling Pathways

Keap1/Nrf2/HO-1 axis is an important regulator of the endogenous antioxidant response associated with oxidative stress and ALI [32,33]. The LPS-induced group alone decreased the protein expression of catalase, SOD, GPx (Figure 5A) and Nrf2, and elevated that of Keap1, compared with the control group (Figure 5B). Moreover, the expression of the antioxidant enzymes HO-1 and Nrf2 could be increased by pretreating the mice with hispolon and Dex.

Peroxisome proliferator-activated receptor γ (PPARγ) is a group of nuclear hormone receptors whose activation has anti-inflammatory effects in many inflammatory disease models, including ALI [34]. As shown in Figure 5B, PPARγ was reduced in lungs with LPS-induced ALI, while pretreatment with hispolon was found to increase PPARγ activity said tissues. This study demonstrated that hispolon increases the relative expression levels of antioxidative proteins following challenges with LPS.

### 3.8. Suppression of TLR4/PI3K/Akt/mTOR Axis

TLR signaling triggers the release of inflammatory cytokines in LPS-induced ALI. The PI3K/Akt/mTOR signaling pathway controls cell growth and oxidative stress in pulmonary inflammation. As Figure 5C shows, the LPS-only group demonstrated an increase the expression of the TLR4, PI3K, Akt and mTOR proteins compared to the control. Hispolon treatment inhibited the expression of this signal transduction pathway compared to the LPS challenge group. These data highlight that hispolon confers protection by inhibiting the TLR4/PI3K/Akt/mTOR signal following an LPS challenge.

### 3.9. Hispolon Reduces ER Stress responses and autophagy inhibition

To investigate the activation of ER stress after LPS challenge in mice, we monitored the expression of six ER stress markers—PERK, IRE1 ATF6, GRP94, CHOP and caspase 12. As shown in Figure 6A, PERK, IRE1 ATF6, GRP94, CHOP and caspase 12 were upregulated by LPS-induced ALI. The results indicate that hispolon inhibited this increase in the expression of ER stress related proteins following LPS challenge.

Autophagy is a physiologically stable regulator of cells in neutrophil-mediated inflammation [35]. To demonstrate whether autophagy is involved in LPS challenge, ALI, and the influence of hispolon on autophagy, we investigated the LC3-II/I and Beclin 1 protein expression, widespread signaling markers of autophagosomes. Preliminary data indicated that their levels were raised in the LPS-challenge group, significantly. However, hispolon pretreatment decreased their levels compared to the LPS alone (Figure 6B). The data showed that hispolon can inhibit ER stress and further promote the activation of the autophagy that interacts to affect the pathological process of LPS-induced ALI.

### 3.10. Hispolon Decreases LKB1/CaMKK–AMPK Signaling

ER stress disrupts the steady state concentration of Ca^2+^ in the ER, causing Ca^2+^ to leak into the cytoplasm. When the Ca^2+^ concentration increases, autophagy is stimulated by Ca^2+^/calmodulin-dependent kinase kinase β (CaMKKβ) and the activation of AMP-activated protein kinase (AMPK). This LKB1/CaMKK–AMPK axis may increase and damage the lung tissue [36]. The p-LKB1, p-CaMKKβ and p-AMPK was decreased after the LPS challenge (Figure 6C). Hispolon treatment elevated the phosphorylation of LKB1, CaMKKβ and AMPK expression compared to the LPS challenge ALI. Our experiments consistently showed that hispolon prevented ER Ca^2+^ leakage by modulating LKB1 and CaMKKβ and, thereby, the ER stress mediated activation of AMPK and autophagy in LPS challenge ALI.

### 3.11. Downregulation of Apoptosis-Related Proteins

Apoptosis is related to protein expression changes of the Bax, Bcl-2 and caspase-3 [37]. According to the results shown in Figure 6D, the LPS-treated group exhibited a low expression of Bcl-2 and the expression level of Bax and caspase-3 proteins were high compared to the control. The data also show that hispolon treatment increased the level of Bcl-2 and decreased that of Bax and caspase-3 compared to the LPS-induced ALI group. These data reveal that hispolon plays a pivotal function of anti-apoptosis in LPS-induced ALI mice.

### 3.12. Hispolon and NAC Reduce the Expression of the AMPK and Nrf-2 Proteins

ROS participate in a variety of signal-linked responses that can promote inflammation and tissue damage. Thus, we examined whether hispolon inhibited LPS-induced ROS production in mice. ROS levels increased after LPS stimulation, while hispolon decreased ROS levels (Figure 7A). However, research results indicate that hispolon may be related to the properties of antioxidants in LPS-challenged mice.

To determine whether a ROS inhibitor (*N*-acetyl-l-cysteine; NAC) could suppress the antioxidant effects, we treated LPS-challenged mice with NAC (150 mg/kg), to investigate the patterns of the protein expression related to the AMPK and Nrf-2 proteins. LPS-induced ROS release was suppressed by the ROS inhibitor. Additionally, compared to the LPS alone, the phosphorylation of the AMPK protein was inhibited and the Nrf-2 protein was increased by co-treatment with hispolon and NAC (Figure 7B,C). Furthermore, co-treatment with hispolon and NAC promoted AMPK phosphorylation and increased HO-1 in the cytosol, while decreasing Nrf-2 expression in the nucleus, significantly. These preliminary results showed that hispolon elevated the activity of Keap1/Nrf2/HO-1 and LKB1/CaMKK-AMPK axis in LPS-challenged ALI mice.

### 3.13. Hispolon and an AKT inhibitor (LY294002) Reduce the Inflammatory Related Proteins

To determine whether an AKT inhibitor (LY294002) could suppress the inflammatory effects, we investigated the patterns of the protein expression related to the TLR4/PI3K/Akt/mTOR pathway in LPS-challenged mice treated with hispolon or LY294002. As displayed in Figure 8A, the iNOS, COX-2, p-AKT and nuclear NF-κB levels were dramatically increased after LPS stimulation. However, co-treatment with hispolon or LY294002 significantly decreased LPS-challenged iNOS, COX-2, p-AKT and nuclear NF-κB expressions in the lung tissues. Thus, these results show that hispolon decreased the inflammatory effects and the activity of TLR4/PI3K/Akt/mTOR pathway in LPS-challenged ALI mice.

### 3.14. Hispolon and an ER Stress Inhibitor Reduces the Expression of Inflammatory and ER Stress Related Proteins

The patterns of protein expression related to the inflammatory and ER stress signal transduction pathways in LPS-challenged mice treated with hispolon or 4-PBA were investigated, to determine if an ER stress inhibitor (4-phenyl butyric acid; 4-PBA) could suppress them. As shown in Figure 8B, the iNOS, COX-2, Beclin-1, caspase 12 and CHOP levels were dramatically increased after LPS stimulation. Furthermore, co-treatment with hispolon or 4-PBA significantly decreased iNOS, COX-2, Beclin-1, caspase-12 and CHOP in lung tissues (Figure 8B). This indicates that hispolon decreased the activity of the inflammatory and ER stress signal transduction pathways in LPS-challenged ALI mice.

## 4. Discussion

ALI is a severe lung disease and there is no good medicine to treat ALI clinically, so the development of ALI treatment strategies is urgently needed [2]. LPS is considered to be one of the factors leading to ALI, which causes fluid to enter the lung tissue and increases microvascular permeability, causing the symptoms of acute inflammation in the tissue [2,3]. LPS-induced ALI mouse models have been shown to be useful models for studying potential treatments against human ALI. The current study utilized an in vivo mouse model to study the protective effects of hispolon in LPS challenge ALI. The present study found that LPS induces neutrophil infiltration and pulmonary edema, and that treatment with hispolon significantly reduces these effects. Oxidative stress induces the activation of macrophages and lipid oxidation, leading to pro-inflammatory cytokine expression. The study also demonstrates that, in the first line of defense against infection, these stress markers ultimately release various pro-inflammatory mediators and recruit neutrophils. We measured the oxidative stress marker antioxidant enzymes and MPO. Surprisingly, hispolon significantly improved the oxidative stress conditions and decreased the pro-inflammatory cytokine expression caused by LPS. Thus, we discovered that hispolon treatment decreased LPS-mediated lung edema, neutrophil infiltration and inflammatory cytokine secretion in the BALF, while controlling the TLR4/PI3K/Akt/mTOR, Keap1/Nrf2/HO-1 pathway and suppressing oxidative stress and the ER stress signaling pathways. In short, the in vivo data indicate that hispolon has anti-inflammatory and antioxidant functions that prevent the lung damage caused by LPS.

Hispolon is an active polyphenolic compound. There is increasing evidence in the literature that hispolon has a wide range of medicinal properties, such as antioxidant, anti-inflammatory, immunomodulatory, antiviral, hepatoprotective and anticancer activities. Among them, the antitumor activity of hispolon has been detailed in different studies, and it has been observed to inhibit the growth of cancer cells by inducing cell cycle arrest, apoptosis and metastasis inhibition. [25,26,27,28,29]. However, the toxicity of hispolon that causes these effects is not fully understood. A recent study has revealed that hispolon showed no significant toxicity for phorbol ester (TPA)-treated and untreated MDA-MB-231 cells between 0 and 40 µM (8.8 µg/mL) for 24 h, thus avoiding anti-proliferation due to hispolon interference caused by activity [38]. Hispolon, with a concentration of 0 to 20 µM (0–4.4 µg/mL), induces the expression of HO-1 protein in BV-2 cells for 24 h under the stimulation of LPS or lipoteichoic acid (LTA), thereby effectively inhibiting the production of iNOS/NO and having no cytotoxicity [29]. Hispolon was also shown to be less cytotoxic to normal cells [39]. Furthermore, hispolon induces apoptosis in acute myeloid leukemia cells and inhibits AML xenograft tumor growth in vivo, where the high dose (10 mg/kg) used is the same as in this study [40].

TLR4, a member of the Toll-like receptors, localizes to the cell membrane and cytoplasm and is a pattern recognition receptor for LPS. LPS-activated macrophages trigger a signaling cascade by releasing inflammatory mediators, resulting in multiple organ damage including ALI [30,36]. Pro-inflammatory cytokines trigger macrophage expression by activating the NF-κB pathway in inflammatory diseases [37]. In ALI patients, an increase in pro-inflammatory cytokines has been observed and are associated with major inflammatory disorders [4]. In the current study, the administration of hispolon significantly ameliorated pulmonary inflammation, hemorrhage and the thickening of the alveolar septa. Furthermore, hispolon treatment diminished the levels of pro-inflammatory cytokines after LPS challenge. These preliminary results indicate that hispolon plays a favorable inhibitory role in LPS-induced ALI.

NF-κB serves as a signal regulator in inflammation, cell proliferation and differentiation [4]. In clinical trials, the activation of NF-κB was increased in patients in response to the overexpression of NF-κBp65 in alveolar macrophages, under severe bacterial infection [37,41], compared with in the control group. From the above experiments, we speculate that hispolon could increase IκBα during LPS challenge to suppress NF-κB, with the suppression of IKK phosphorylation in the cytosol, preventing NF-κB p65 nuclear translocation.

The MAPK (ERK1/2, p38MAPK, and JNK) cascade has been shown to play an extracellular signal transduction, such as LPS-induced inflammatory cytokine production [8,32]. The defective activation or pharmacological inhibition of ERK or p38 reduces the production of LPS-induced pro-inflammatory cytokines [37]. Thus, targeting TLR4 or its downstream MAPK signaling may prevent the abnormal immune response associated with ALI. In addition, hispolon activated ERK1/2, p38MAPK, and JNK phosphorylation in cancer cells, such as nasopharyngeal carcinomas cells [42], cervical cancer cells [43] and hepatocellular carcinoma cells [27]. MAPK signaling plays a critical role in the chemotherapy drugs and metastasis [44,45]. Furthermore, LPS is presented to the TLR4/MD2 complex via LPS binding protein and CD14. The formation of the TLR4/MD2/LPS complex causes the phosphorylation of MAPK and activates downstream NF-κB [46]. Both TLR4/MD2 and MAPK/NF-κB are required for LPS-induced inflammation. Therefore, many natural compounds inhibit MAPK/NF-κB signaling in a variety of ways, showing the treatment or relief of lung inflammation. This indicates that it may be the target of acute inflammatory drug treatment. In addition, the inhibition of the MAPK pathway can reduce the transcription and oxidative stress of proinflammatory mediators [47]. In this study, hispolon suppressed MAPK phosphorylation and decreased the pro-inflammatory cytokines expressions via NF-κB activation in the LPS-induced model. Thus, hispolon significantly prevented the degradation NF-κB and IκBα and the phosphorylation of MAPK in LPS-induced ALI mice.

Recent reports suggest that the PI3K/Akt/mTOR axis is a key point for TLRs/NF-κB to coordinate inflammatory responses [5,28]. It is clear that the activation of TLR leads to the recruitment of PI3K, by allowing adaptor molecules to enter the receptor complex. Thus, PI3K/AKT/mTOR signaling regulates cell growth, proliferation, migration, invasion, survival, apoptosis and autophagy, and is tightly regulated in the TLR signaling pathway [30]. This shows that the LPS-induced inflammatory reaction was mediated through the TLR4 receptor, which improved the PI3K, p-Akt, and p-mTOR protein levels. Pre-treatment with hispolon significantly decreased PI3K, p-Akt and p-mTOR levels after the LPS challenge. Thus, the PI3K/AKT/mTOR axis is a potential predictor for hispolon treatment in the LPS-challenged mice. It is proven by the above experiments that the PI3K/Akt/mTOR and TLR/NF-ĸB pathways are coordinated in regulating the inflammatory response triggered by LPS.

LPS significantly enhances ROS production, promotes inflammatory responses, decreases antioxidant enzyme (catalase, SOD and GPx) activity, and increases the MPO activity associated with neutrophil infiltration in the development of ALI [30,36]. However, appropriate ROS levels help to protect humans from external stimuli or pathological damage. By contrast, the excessive production of ROS is thought to induce cell damage and oxidative stress. The data of this study indicate that hispolon treatment diminished the production of ROS, increased Nrf-2 expression in the nucleus and increased the phosphorylation of AMPK after LPS challenge. In addition, cells contain a variety of antioxidant enzymes to prevent cell damage by reducing oxidative stress [30]. Our preliminary data showed that hispolon increases the expression of the antioxidant proteins in the ALI model.

In the presence of oxidative stress, the release of Nrf2 from Keap1 was activated. Nrf2 is involved in the induction of HO-1 and GP, which can eliminate ROS through oxidative damage [7,10]. HO-1 may suppress oxidative stress through the activation of the PI3K/Akt or Nrf2 axis in sepsis-induced ALI [4]. Of interest, HO-1 can prevent liver and endothelial cell apoptosis mediated by ER stress in diabetic animals [48]. These results have suggested that the special protective effects of hispolon were mediated by inhibiting oxidative stress via the Keap1/Nrf2/HO-1 axis and reducing ROS production.

The ER stress response is associated with several liver diseases, for example, ALI, obesity-related fatty liver and viral hepatitis [11,12]. Studies have shown that ER stress is a key strength of mediation in the LPS-induced inflammation and the protective effect of 4-PBA is related to ER stress and autophagy in LPS-challenged mice. Moreover, 4-PBA is a chemical chaperone, currently undergoing clinical trials, that inhibits ER stress, and is promising as a future research candidate for new asthma therapies [49]. As shown in the results section, p-PERK, GRP78 and CHOP expression was induced by LPS. Hispolon treatment decreased the levels of these, and ER sensor proteins, such as ATF6, caspase-12 and IRE1, can inhibit their activation. These findings indicate that hispolon plays an anti-inflammatory role by ameliorating ER stress.

The activation of autophagy can degrade proteins against ER stress induced toxicity [18,19]. Reports have indicated that the ER stress-activated PERK protein may induce LC3, which is involved in the induction of autophagy. In the current study, the upregulation of autophagy-related proteins such as LC3-II and Beclin 1 was observed in lung tissue after 6 h of induction by the intratracheal instillation of LPS. This could suggest that autophagy acts as a compensatory mechanism during lung injury, and the induction of autophagy may improve lung cell survival by providing energy in adverse environments. To date, the exact role of the autophagy process in ALI remains unclear. Specifically, ER stress could be an upstream mediator to regulate autophagy. In addition, autophagy is a multi-faceted process, and changes in autophagy signaling are often observed in cancer and the recruitment of LC3-II-mediated of phagocytic membrane to damaged organelles. Thus, hispolon activated LC3-II in nasopharyngeal carcinomas cells [37]. Our preliminary data showed that hispolon activated the TLR4/PI3K/Akt/mTOR axis and the protein expression of the LC3-II and Beclin 1 was reduced, suggesting that ER stress is the cause of LPS challenge autophagy activation in mice.

Apoptosis is induced when cells receive internal or external signals. Intrinsic apoptosis is caused by a variety of disturbances to the microenvironment, including DNA damage and ER and ROS stress. The apoptosis induced by ER stress is an example of intrinsic apoptosis signaling [21]. ER stress and inflammatory signaling are intrinsically linked through a variety of mechanisms, and ER stress increases the induction of inflammatory cytokines in macrophages by LPS by several times [50,51]. Our data showed that hispolon suppressed Bcl-2 protein expression and increased Bax and caspase-3 protein expression, well-known apoptosis markers, resulting in the inhibition of severe ER stress and limiting the lung injury triggered by lung cell apoptosis and liver inflammation in ALI mice induced by LPS. In addition, the regulating caspase activation and inhibition mechanisms is a key factor in the treatment of cancer, because these processes can induce apoptosis [52]. Hispolon induces cancer cells apoptosis through the activation of caspases-8, -9 and -3 and the cleavage of PARP [40,42,43,53]. Moreover, the results of this study indicate that autophagy and apoptosis may play different roles at different stages of LPS-induced ALI. Autophagy peaked at 2 h. However, apoptosis reached its maximal level at later stages (6 h) [54]. The relationship between autophagy and apoptosis can be very complex. Both autophagy and the apoptosis of lung cells can be triggered by extracellular stimulation. Some intracellular signaling pathways, including MAPK and NF-κB, are involved in both types of cell death. A recent study revealed the interaction between autophagy and apoptosis-related intracellular signaling pathways in lung cells, suggesting that autophagy may be a survival mechanism against Fas-mediated apoptosis [55,56]. In this study, hispolon reduced the protein level of Bax and increased the protein level of Bcl-2, resulting in a decrease in the ratio of Bax to Bcl-2 and reduced apoptosis. Further research is needed to study the possible role of autophagy and apoptosis in ALI.

## 5. Conclusions

Our findings suggest that hispolon shows an enhanced therapeutic effect in LPS challenge ALI, via directly inhibiting the inflammatory response. This study reveals the following mechanisms of action: the inhibition of pathological lung changes, the inhibition of lung edema, the inhibition of macrophage tissue infiltration, and inhibit the secretion of various pro-inflammatory cytokines. Together, the research data show that hispolon has potent anti-inflammatory properties via the reduction of iNOS and COX-2 expression, suppression of the NF-κB and MAPK axis, inhibition of the TLR4/PI3K/Akt/mTOR and Keap1/Nrf2/HO-1 signaling pathways and the suppression of oxidative stress and ER stress-induced apoptosis and autophagy (Figure 9). Therefore, hispolon has powerful anti-inflammatory properties and can be developed as an effective therapeutic agent specifically for ALI.

## Figures and Tables

**Figure 1 nutrients-12-01742-f001:**
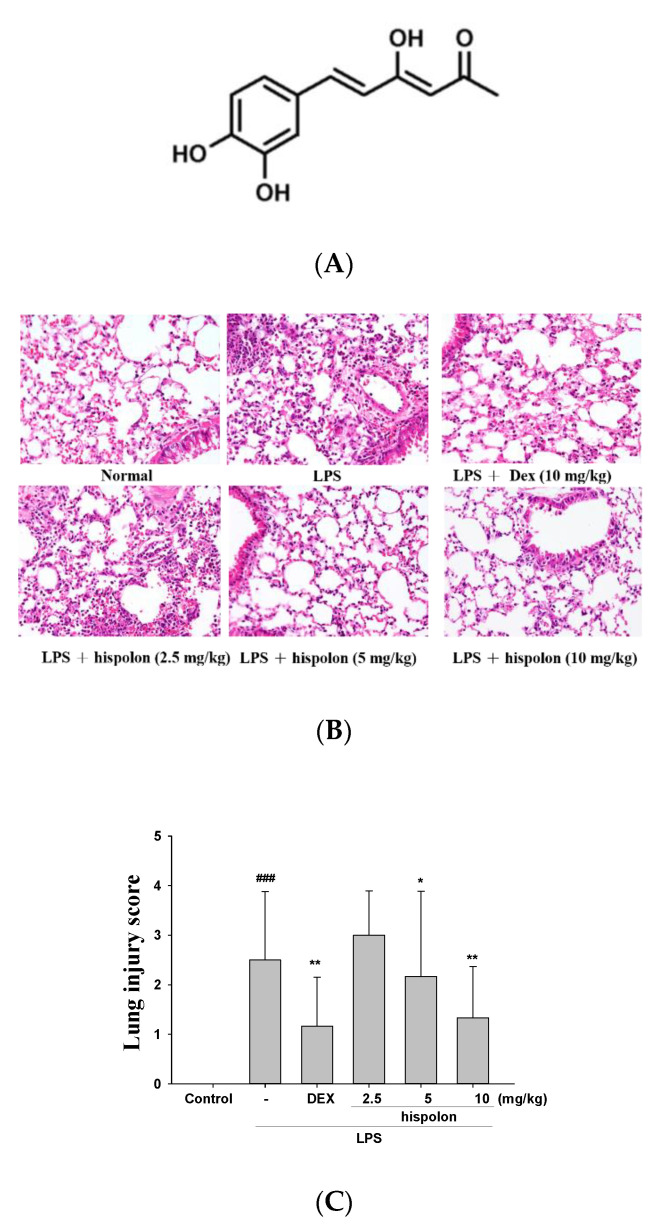
(**A**) The chemical structural formula of hispolon, (**B**) the lung injury scores and (**C**) the effects of hispolon (2.5, 5 and 10 mg/mL) on LPS-induced histopathologic alterations in lung tissues of mice. After LPS challenge, the lungs were prepared for histological assessment. Sections were stained with H&E and viewed under magnification (400×). Data are presented as the means ± S.E.M (n = 6). ^###^
*p* < 0.001 versus the control group. * *p* < 0.5 and ** *p* < 0.01 versus the LPS group.

**Figure 2 nutrients-12-01742-f002:**
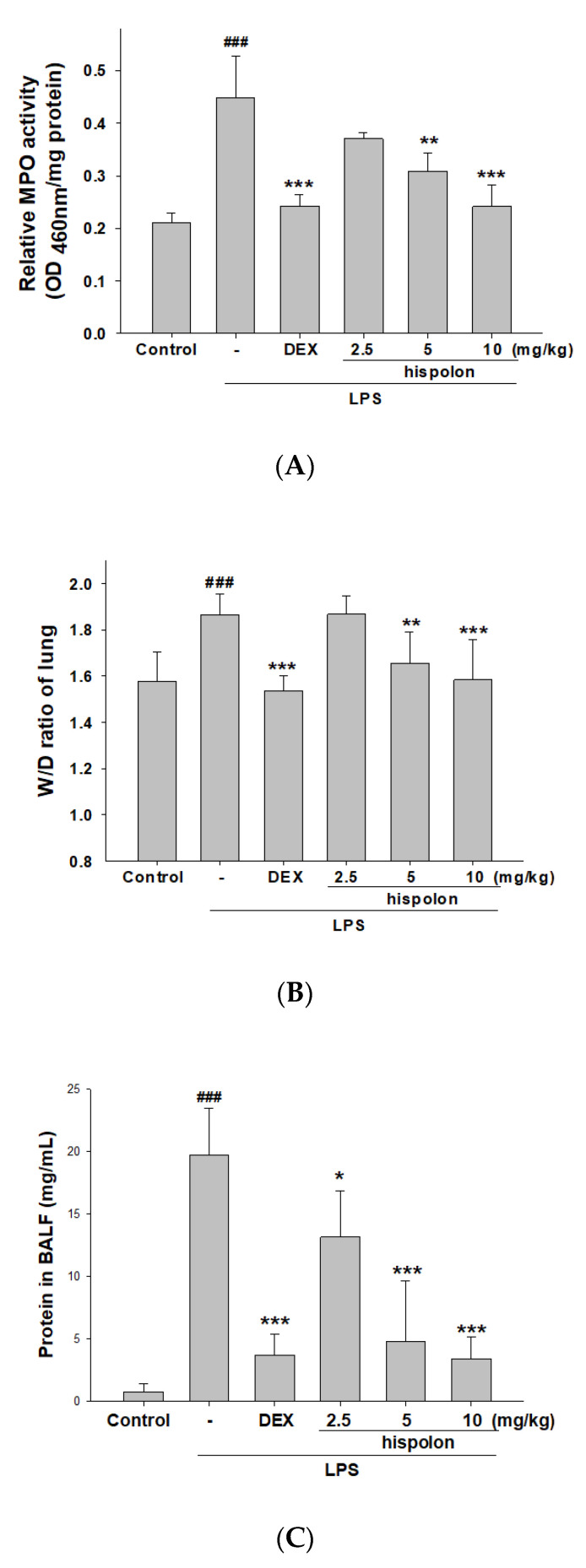
Hispolon improved (**A**) pulmonary edema (W/D ratio) and (**B**) Myeloperoxidase (MPO) activity, and decreased (**C**) cell counts and (**D**) total protein in the bronchoalveolar lavage fluid (BALF). Lung tissues were measured by calculating the W/D ratios. Total cells and total proteins of BALF were assessed. Data are presented as means ± S.E.M. (n = 6). ^###^
*p* < 0.001 versus the control group. * *p* < 0.05, ** *p* < 0.01 and *** *p* < 0.001 versus the lipopolysaccharide (LPS) group.

**Figure 3 nutrients-12-01742-f003:**
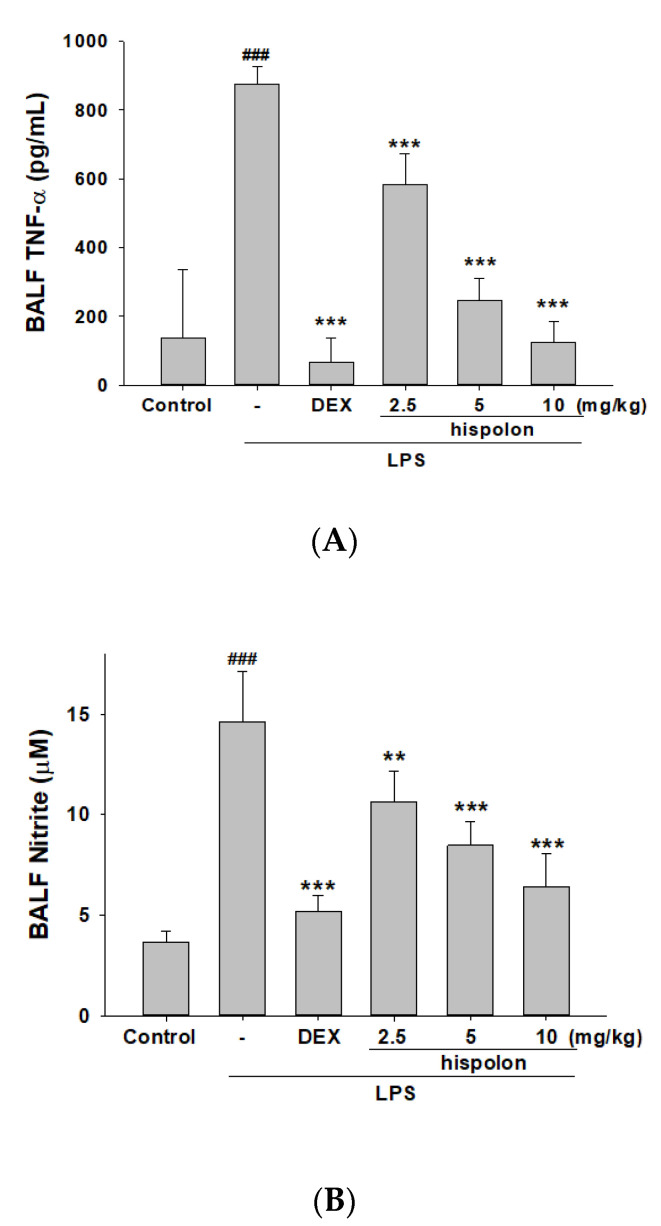
Hispolon decreased (**A**) NO, (**B**) TNF-α, (**C**) IL-1β and (**D**) IL-6 in the BALF. Pro-Inflammatory cytokine were measured after LPS challenge by ELISA. Data are represented as means ± S.E.M. (n = 6). ^###^
*p* < 0.001 versus the control group. ** *p* < 0.01 and *** *p* < 0.001 versus the LPS group.

**Figure 4 nutrients-12-01742-f004:**
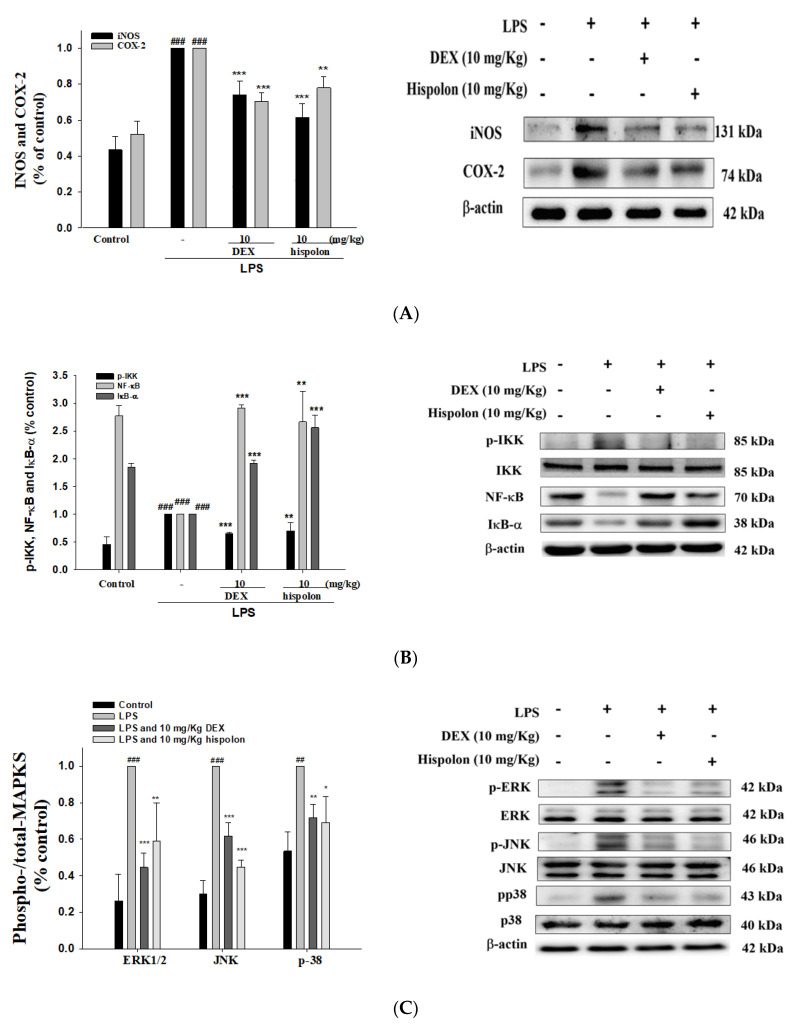
Effects of hispolon on the activation of (**A**) iNOS, COX-2, (**B**) IκB-α, NF-κB and (**C**) the phosphorylation of MAPK axis in lung tissue of LPS-induced acute lung injury (ALI) mice. Lung tissue extracts were subjected to Western blot analysis using antibodies for iNOS, COX-2, IκB-α and NF-κB and MAPK phosphorylation. Data are represented as means ± S.E.M. (n = 3). ^###^
*p* < 0.001 versus the control group. * *p* < 0.05, ** *p* < 0.01 and *** *p* < 0.001 versus the LPS group.

**Figure 5 nutrients-12-01742-f005:**
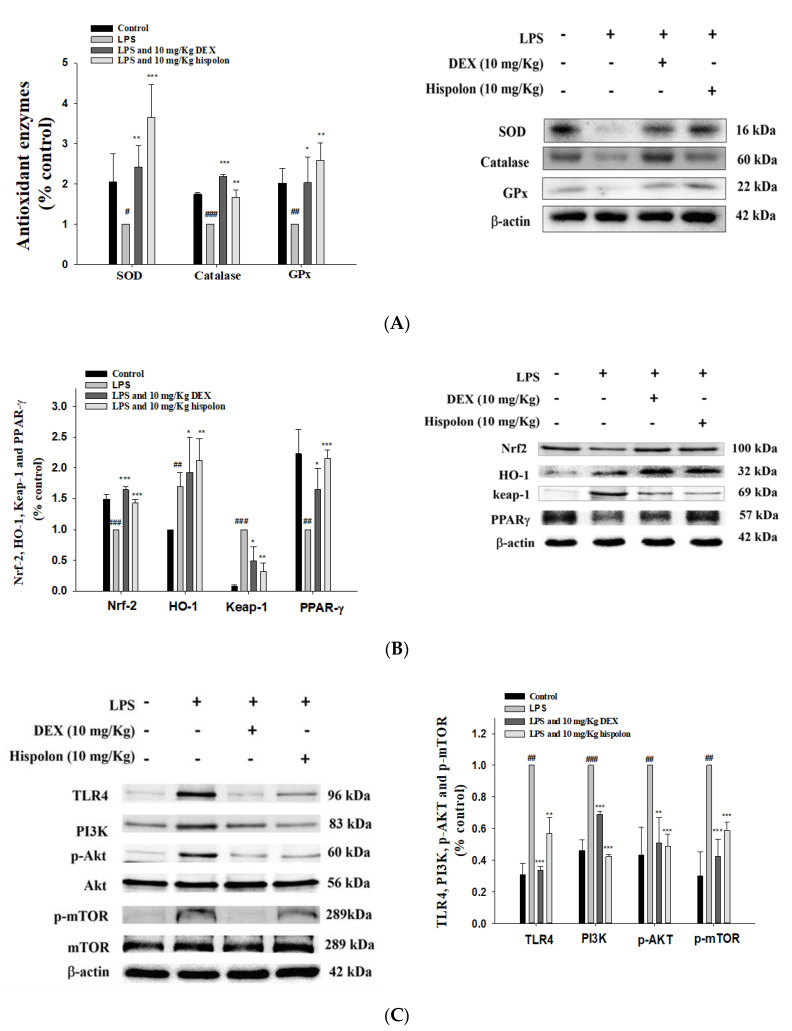
Effects of hispolon on (**A**) LPS challenge catalase, superoxide dismutase (SOD) and GPx; (**B**) HO-1, Nrf2, Keap1 and PPARγ; (**C**) and TLR4, PI3K, Akt and mTOR protein expression in the lungs. Lung tissue extracts were subjected to Western blot analysis using antibodies for catalase, SOD, GPx, HO-1, Nrf2, Keap1, PPARγ, TLR4, PI3K, Akt and mTOR. Data are represented as means ± S.E.M. (n = 3). ^##^
*p* < 0.01 and ^###^
*p* < 0.001 versus the control group. * *p* < 0.05, ** *p* < 0.01 and *** *p* < 0.001 versus the LPS group.

**Figure 6 nutrients-12-01742-f006:**
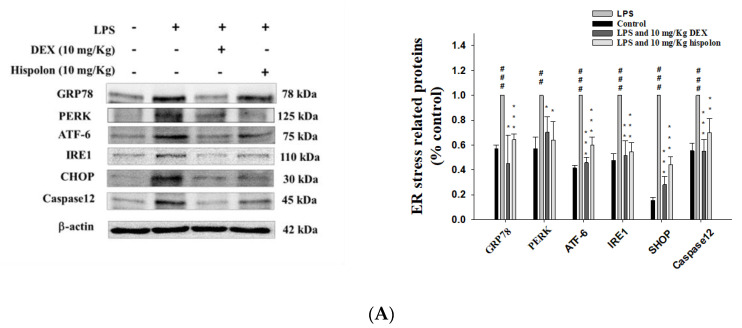
Effects of hispolon on (**A**) ER stress relative enzymes (PERK, IRE1 ATF6, GRP94, CHOP and caspase-12), (**B**) autophagy related enzymes (LC3-II/I and Beclin 1), (**C**) LKB1/CaMKK–AMPK signaling, and (**D**) apoptosis-related protein (Bcl-2, Bax and caspase-3) levels in the lungs of LPS challenge ALI. Lung tissue extracts were subjected to Western blot analysis using antibodies for PERK, IRE1 ATF6, GRP94, CHOP, caspase-12, LC3-II/I, Beclin 1, LKB1, CaMKK, AMPK, Bcl-2, Bax and caspase-3. Data are represented as means ± S.E.M. (n = 3). ^##^
*p* < 0.01 and ^###^
*p* < 0.001 versus the control group. * *p* < 0.05, ** *p* < 0.01 and *** *p* < 0.001 versus the LPS group.

**Figure 7 nutrients-12-01742-f007:**
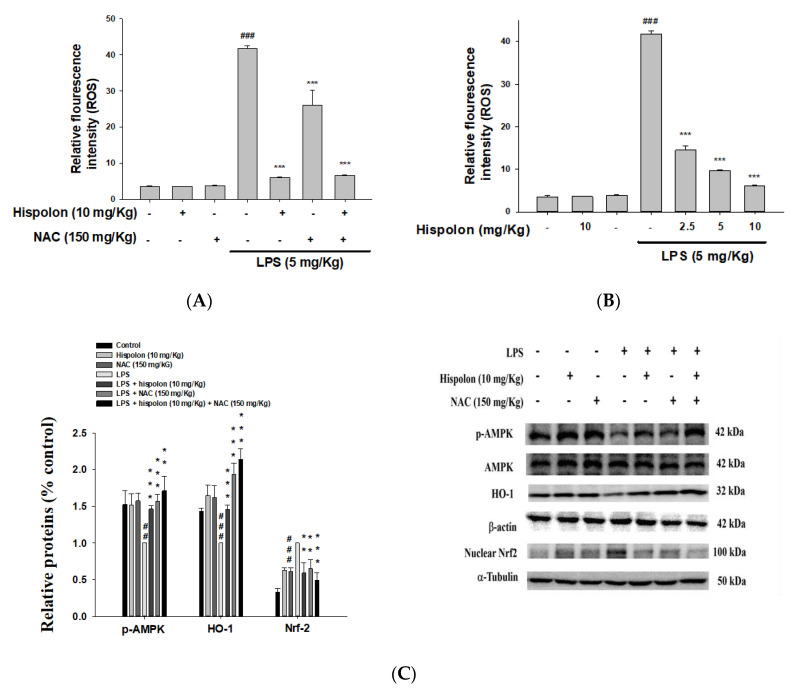
Hispolon downregulated ROS (**A**), a ROS inhibitor (NAC) reduced ROS (**B**) in the BALF and regulated p-AMPK, HO-1 and nuclear Nrf2 (**C**) protein expression in the lungs. ROS was detected after LPS challenge by ELISA. Lung tissue extracts were subjected to Western blot analysis using antibodies for p-AMPK, HO-1 and nuclear Nrf2. Data are represented as means ± S.E.M. (n = 3). ^##^
*p* < 0.01 and ^###^
*p* < 0.001 versus the control group. ** *p* < 0.01 and *** *p* < 0.001 versus the LPS group.

**Figure 8 nutrients-12-01742-f008:**
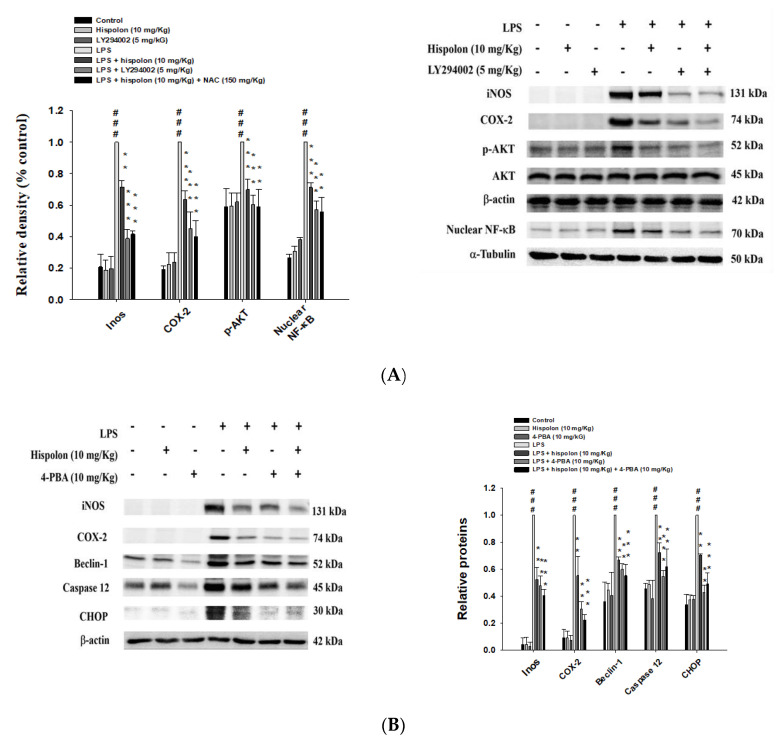
Hispolon, an AKT inhibitor (LY294002) (**A**) and an ER inhibitor (4-PBA) (**B**) regulated protein expression in LPS-challenged mice. Lung tissue extracts were subjected to Western blot analysis, using antibodies for iNOS, COX-2, Beclin-1, caspase 12 and CHOP. Data are represented as means ± S.E.M. (n = 3). ^###^
*p* < 0.001, compared with the control group. ** *p* < 0.01 and *** *p* < 0.001 versus the LPS group.

**Figure 9 nutrients-12-01742-f009:**
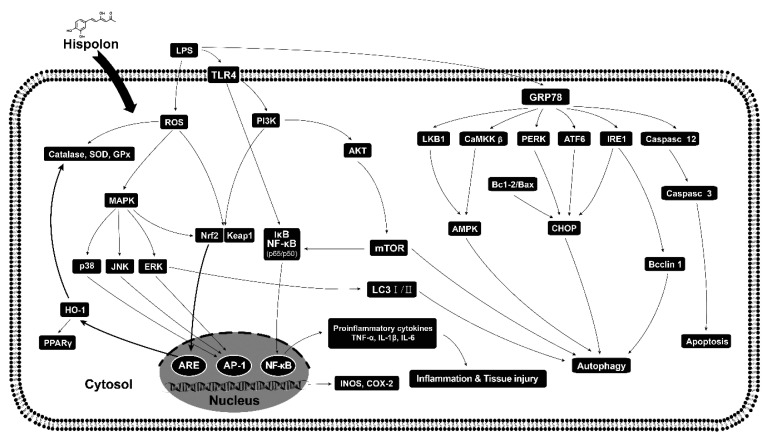
A scheme for the protective effect of hispolon against LPS challenge inflammation.

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
