# Peer review of "Attenuation of Lipopolysaccharide-Induced Acute Lung Injury by Hispolon in Mice, Through Regulating the TLR4/PI3K/Akt/mTOR and Keap1/Nrf2/HO-1 Pathways, and Suppressing Oxidative Stress-Mediated ER Stress-Induced Apoptosis and Autophagy"

_nutrients, 2020, doi:10.3390/nu12061742_

Round 1

Reviewer 1 Report

This is a study reporting the therapeutic effect of Hispolon in a mouse model of LPS-induced acute lung injury. The authors studied the signaling pathways underlying the beneficial effect of HIspolon. The study is very comprehensive, exploring multiple pathways. Previous studies have reported the antioxidant and anti-inflammatory effects of hispolon in other models, notably the same group in a rat model of CCl4-induced liver toxicity. The authors attempted to identify other pathways relevant to that effect. In light of the existing literature, there are some drawbacks to the study that need to be addressed and discussed. The authors do not mention other hispolon studies contradicting their findings. Their results need to be discussed and explained in light of previous findings.

Except for the study by the same authors in rats, most other studies use significantly lower doses of hispolon. This requires a justification of such high dosage in light of the cytotoxicity of the compound at much lower concentrations compared to those used in this study (Chethna et al., 2018; Yun et al., 2019; Chen et al., 2014; Hsiao et al., 2013). In addition, hispolon’s solubility in water or saline appears to be very low. Was it administered in DMSO/ In that case was a similar DMSO concentration used in controls?

The experimental protocol is unclear. It mentions that LPS was administered in sterile saline IP. However it seems like the model is intratracheal administration. Also were the control animals administered intratracheal saline or were they left with no intervention?

The protocol mentions that 6 animals were treated per condition. However the number of animals (n=?) in the legends is confusing. The histological data was obtained from 5 left lungs / per condition then all other lungs had to be used for the rest of the determinations. Were the same lungs used for western blot and for MPO analysis?

In the results:

Paragraph 3.1: the authors state “reduced infiltration of inflammatory cells” This statement needs to be removed, inflammatory cells were neither stained nor quantified. In addition, in the legend of figure 1, the authors state that the red arrows indicate bleeding and inflammatory cells infiltration. However these cannot be seen on this figure.

Paragraph 3.6 : The authors show that Hispolon significantly decreases ERK, p38 and JNK phosphorylation. This is in contradiction with multiple studies showing that hispolon increases ERK, p38 and JNK phosphorylation (Hsieh et al., 2014 and 2018; Hsin etal., 2017; as well as the authors’ study in cells Huang et al., 2011) This discrepancy needs to be discussed.

Paragraph 3.9: Similar to the comments about paragraph 3.6, the authors ignored studies reporting that hispolon induces autophagy with increased LC3II expression (Hsin et al., 2017). Please discuss.

Paragraph 3.11: Similar to the comments about paragraph 3.6 and 3.9, the authors ignored multiple studies including their own 2011 study, reporting that hispolon induces apoptosis (Hsiao et al., 2013; Hsieh et al., 2014; Chen et al., 2014; Yun et al., 2019; . These other studies show that hispolon induces increases in pro-apoptotic Bax and Bim proteins, increased caspase 3 and PARP cleavage and decreased in anti-apoptotic proteins Bcl2 and Bcl-xL, clearly suggesting a pro-apoptotic effect of hispolon, and contrary to the anti-apoptotic effect reported in the current study. These need again to be discussed.

Author Response

Point-by-point Response to the Reviewers

Reviewers' 1 comments:

  This is a study reporting the therapeutic effect of Hispolon in a mouse model of LPS-induced acute lung injury. The authors studied the signaling pathways underlying the beneficial effect of Hispolon. The study is very comprehensive, exploring multiple pathways. Previous studies have reported the antioxidant and anti-inflammatory effects of hispolon in other models, notably the same group in a rat model of CCl4-induced liver toxicity. The authors attempted to identify other pathways relevant to that effect. In light of the existing literature, there are some drawbacks to the study that need to be addressed and discussed. The authors do not mention other hispolon studies contradicting their findings. Their results need to be discussed and explained in light of previous findings.

Except for the study by the same authors in rats, most other studies use significantly lower doses of hispolon. This requires a justification of such high dosage in light of the cytotoxicity of the compound at much lower concentrations compared to those used in this study (Chethna et al., 2018; Yun et al., 2019; Chen et al., 2014; Hsiao et al., 2013). In addition, hispolon’s solubility in water or saline appears to be very low. Was it administered in DMSO/ In that case was a similar DMSO concentration used in controls?

[Reply]: Thank you for your advice. Hispolon is one of the most important polyphenols in S. sanghuang with anti-inflammatory and antitumor effects [1, 2].   In this paper, the animals received hispolon (2.5, 5 and 10 mg/kg) by injected intraperitoneally 1 h prior to LPS administration. The low-dose effect we used in the pre-experiment was not as expected, so we used the current dose. From the low dose (2.5 mg / Kg) of this experiment in the Figure 1, 2 and 3, we can see that its efficacy is not as expected. In addition, the same of animal experiment, hispolon induces apoptosis in acute myeloid leukemia cells and inhibits AML xenograft tumor growth in vivo (Hsiao et al., 2013) [3] that the high dose (10 mg/kg) used is the same as in this study. And, as the review said that hispolon’s solubility in water or saline appears to be very low. Thus, Hispolon (2.5, 5 and 10 mg/kg) suspended in 0.5% CMC solution [4] and injected i.p. We more described it in the material and methods, on the line 137.

Reference:

  1. J. Huang, J.S. Deng, C.S. Chiu, et al., “Hispolon protects against acute liver damage in the rat by inhibiting lipid peroxidation, proinflammatory cytokine, and oxidative stress and downregulating the expressions of iNOS, COX-2, and MMP-9,” Evidence-Based Complementary and Alternative Medicine Research, vol. 2012, pp. 480714, 2012.
  2. C. Chen, H.Y. Chang, J.S. Deng, et al., “Hispolon from Phellinus linteus induces G0/G1 cell cycle arrest and apoptosis in NB4 human leukaemia cells,” The American Journal of Chinese Medicine, vol: 41, pp: 1439–1457, 2013.
  3. P.C. Hsiao, Y.H. Hsieh, J.M. Chow, S.F. Yanget al., “Hispolon induces apoptosis through JNK1/2-mediated activation of a caspase-8, -9, and -3-dependent pathway in acute myeloid leukemia (AML) cells and inhibits AML xenograft tumor growth in vivo,”Journal of Agricultural and Food Chemistry,  61, pp. 10063-73, 2013. 
  4. S El-Agamy, “Nilotinib Ameliorates Lipopolysaccharide-Induced Acute Lung Injury in Rats, “Toxicology and applied pharmacology, vol. 253, pp. 153-160, 2011.

The experimental protocol is unclear. It mentions that LPS was administered in sterile saline IP. However, it seems like the model is intratracheal administration. Also were the control animals administered intratracheal saline or were they left with no intervention?

[Reply]: Thank you for your advice. In this animal model, mice were intratracheally instilled with 5 mg/kg LPS to induce lung injury, while sterile saline was used as the control (1-2). Thus, we more described it in the material and methods, on the line 138-139.

Reference:

  1. Tao, N. Li, Z. Zhang, et al., “Erlotinib protects LPS-induced acute lung injury in mice by inhibiting EGFR/TLR4 signaling pathway,”Shock, vol. 51, pp. 131-138, 2019.
  2. Fu, S.Hao, X. Xu, et al.,"Activation of SIRT1 ameliorates LPS-induced lung injury in mice via decreasing endothelial tight junction permeability," Acta Pharmacologica Sinica. vol. 40, pp. 630-641, 2019.

The protocol mentions that 6 animals were treated per condition. However, the number of animals (n=?) in the legends is confusing. The histological data was obtained from 5 left lungs / per condition then all other lungs had to be used for the rest of the determinations. Were the same lungs used for western blot and for MPO analysis?

[Reply]: Thank you for your advice. In this animal model, mice were randomly divided into six treatment groups (n=6). The histological data was obtained from 5 left lungs / per condition, because the poor data in each group is deleted. In order to reduce the review’s confuse, we recalculate and count the data (n=6) (Fig. 1C). In addition, the lower lobe of right lung tissue was removed for histopathological analysis and the lower of left lung tissues were harvested for calculated as W/D ratio. So, the left lung tissues used for western blot and for MPO analysis.

In the results:

Paragraph 3.1: the authors state “reduced infiltration of inflammatory cells” This statement needs to be removed, inflammatory cells were neither stained nor quantified. In addition, in the legend of figure 1, the authors state that the red arrows indicate bleeding and inflammatory cells infiltration. However, these cannot be seen on this figure.

[Reply]: Thank you for your advice. We deleted this sentences and figure as review suggestion.

Paragraph 3.6 : The authors show that Hispolon significantly decreases ERK, p38 and JNK phosphorylation. This is in contradiction with multiple studies showing that hispolon increases ERK, p38 and JNK phosphorylation (Hsieh et al., 2014 and 2018; Hsin etal., 2017; as well as the authors’ study in cells Huang et al., 2011) This discrepancy needs to be discussed.

[Reply]: Thank you for your advice. we more described it in the discussion, on the line 557-560.

Paragraph 3.9: Similar to the comments about paragraph 3.6, the authors ignored studies reporting that hispolon induces autophagy with increased LC3II expression (Hsin et al., 2017). Please discuss.

[Reply]: Thank you for your advice. we more described it in the discussion, on the line 606-609.

Paragraph 3.11: Similar to the comments about paragraph 3.6 and 3.9, the authors ignored multiple studies including their own 2011 study, reporting that hispolon induces apoptosis (Hsiao et al., 2013; Hsieh et al., 2014; Chen et al., 2014; Yun et al., 2019; . These other studies show that hispolon induces increases in pro-apoptotic Bax and Bim proteins, increased caspase 3 and PARP cleavage and decreased in anti-apoptotic proteins Bcl2 and Bcl-xL, clearly suggesting a pro-apoptotic effect of hispolon, and contrary to the anti-apoptotic effect reported in the current study. These need again to be discussed.

[Reply]: Thank you for your advice. we more described it in the discussion, on the line 621-624.

Reviewer 2 Report

The authors showed the anti-inflammatory effects of hispolon with an interesting mechanistic approach on mice.

This is a very interesting paper but I did not understand the use of dexamethasone compared to hispolon. They used the same concentration of these two compounds and the effects were similar, so why we would prefer hispolon instead of dexamethasone. If there is a clinical implication it is not evincible from the manuscript and the readers could suppose that is the same effect. For this reason, I suggest to stand out the hispolon and to better elucidate the compounds used to compare the effects of hispolon.

In material and methods section, authors should specify how many animals were used for each group.

Author Response

Reviewers' 2 comments:

The authors showed the anti-inflammatory effects of hispolon with an interesting mechanistic approach on mice.

This is a very interesting paper but I did not understand the use of dexamethasone compared to hispolon. They used the same concentration of these two compounds and the effects were similar, so why we would prefer hispolon instead of dexamethasone. If there is a clinical implication, it is not evincible from the manuscript and the readers could suppose that is the same effect. For this reason, I suggest to stand out the hispolon and to better elucidate the compounds used to compare the effects of hispolon.

[Reply]: Thank you for your advice. In this paper, we use dexamethasone to be a positive control. hispolon is a natural bioactive polyphenol from S. sanghuang, Phellinus linteus or Inonotus hispidus used as traditional medicine and functional food in Taiwan, Korea, China and Japan. The purpose of this study was to determine suppressed the activation of the LPS-induced inflammatory pathways, oxidative injury, ER stress, apoptosis, autophagy and has potential therapeutic efficacy in major anterior segment lung diseases.

In material and methods section, authors should specify how many animals were used for each group.

[Reply]: Thank you for your advice. In this paper, we more described it in the material and methods and figure legends section as suggested.

Round 2

Reviewer 1 Report

V

Although the authors now mention other studies contradicting their findings, including their own, they do not discuss why all other studies findings are contradicting the findings in their current studies. As noted before and now stated in the revised discussion, in all other studies Hispolon increases MAP Kinases activity, autophagy and apoptosis, completely contradicting their present findings. These contradictory findings are puzzling and require a discussion of potential reason that would underlie the discrepancies in their present results. Instead the authors just cited the studies. In addition the authors do not discuss hispolon’s toxicity at such high concentration.

Author Response

Point-by-point Response to the Reviewers

Reviewers' 1 comments:

    Although the authors now mention other studies contradicting their findings, including their own, they do not discuss why all other studies findings are contradicting the findings in their current studies. As noted before and now stated in the revised discussion, in all other studies Hispolon increases MAP Kinases activity, autophagy and apoptosis, completely contradicting their present findings. These contradictory findings are puzzling and require a discussion of potential reason that would underlie the discrepancies in their present results. Instead the authors just cited the studies.

[Reply]: Thank you for your advice. We more described it in the discussion, on the line 574-581 and 645-655.

In addition, the authors do not discuss hispolon’s toxicity at such high concentration.

 [Reply]: Thank you for your advice. We more described it in the discussion, on the line 537-550.
